# Variety-Driven Effect of Rhizosphere Microbial-Specific Recruitment on Drought Tolerance of *Medicago ruthenica* (L.)

**DOI:** 10.3390/microorganisms11122851

**Published:** 2023-11-24

**Authors:** Jing Xing, Wenqiang Fan, Jiani Wang, Fengling Shi

**Affiliations:** Key Laboratory of Grassland Resources of the Ministry of Education, Key Laboratory of Forage Cultivation, Processing and High-Efficiency Utilization of the Ministry of Agriculture, College of Grassland, Resources and Environment, Inner Mongolia Agricultural University, Hohhot 010010, China; xjing_wm@126.com (J.X.); fanwinkion@163.com (W.F.); wangjiani123123@163.com (J.W.)

**Keywords:** *M. ruthenica* (L.), different varieties, drought stress, rhizosphere microbes, 16S rRNA sequencing

## Abstract

As one of the environmental factors that seriously affect plant growth and crop production, drought requires an efficient but environmentally neutral approach to mitigate its harm to plants. Soil microbiomes can interact with plants and soil to improve the adverse effects of drought. *Medicago ruthenica* (L.) is an excellent legume forage with strong drought tolerance, but the key role of microbes in fighting drought stress remains unclear. What kind of flora plays a key role? Is the recruitment of such flora related to its genotype? Therefore, we selected three varieties of *M. ruthenica* (L.) for drought treatment, analyzed their growth and development as well as their physiological and biochemical characteristics, and performed 16S rRNA high-throughput sequencing analysis on their rhizosphere soils to clarify the variety-mediated response of rhizosphere bacteria to drought stress. It was found that among the three varieties of *M. ruthenica* (L.), Mengnong No.2, Mengnong No.1 and Zhilixing were subjected to drought stress and showed a reduction in plant height increment of 24.86%, 34.37%, and 31.97% and in fresh weight of 39.19%, 50.22%, and 41.12%, respectively, whereas dry weight was reduced by 23.26%, 26.10%, and 24.49%, respectively. At the same time, we found that the rhizosphere microbial community of Mengnong No. 2 was also less affected by drought, and it was able to maintain the diversity of rhizosphere soil microflora stable after drought stress, while Mennong No. 1 and Zhilixing were affected by drought stress, resulting in a decrease in rhizosphere soil bacterial community diversity indices to 92.92% and 82.27%, respectively. Moreover, the rhizosphere of Mengnon No. 2 was enriched with more nitrogen-fixing bacteria Rhizobium than the other two varieties of *M. ruthenica* (L.), which made it still have a good ability to accumulate aboveground biomass after drought stress. In conclusion, this study proves that the enrichment process of bacteria is closely related to plant genotype, and different varieties enrich different types of bacteria in the rhizosphere to help them adapt to drought stress, and the respective effects are quite different. Our results provide new evidence for the study of bacteria to improve the tolerance of plants to drought stress and lay a foundation for the screening and study mechanism of drought-tolerant bacteria in the future.

## 1. Introduction

Water scarcity affects between 100 and 200 million people globally, especially those in developing regions, which are home to about 70% of the world’s drylands, making their lives very difficult [1]. According to the projected climate change, droughts will become more frequent and severe [2]. Whereas drought is one of the most important factors affecting crop yields, increasingly drastic climate change and growing water scarcity are challenges to global crop production [3].

*Medicago ruthenica* (L.) is widely distributed in northern China and plays an important role in windbreak, sand fixation, soil and water conservation, and natural grassland improvement [4]. Meanwhile, it is characterized by barrenness tolerance, trampling resistance, and outstanding cold and drought resistance, which can provide excellent genetic resources for pasture grass [5]. Its high nutritional value and good palatability make it an important forage legume [6]. In summary, *M. ruthenica* (L.) has high ecological value, scientific research value, and economic value.

Previous studies on drought tolerance in *M. ruthenica* (L.) have mostly focused on morphology, physiology, and genomics [7,8,9,10]. However, there are many problems in improving drought tolerance in lentils: traditional breeding methods are time-consuming and labor-intensive, and mutations brought about by mutation breeding are full of uncertainties. The unstable expression of drought-resistant transgenic crops and the public acceptance of transgenic crops have all affected the process of drought-resistant *M. ruthenica* (L.) breeding and application. In contrast, positive effects from plant–microbe interactions are gradually becoming a viable pathway for drought tolerance [11,12], which is a more effective approach than plant breeding and genetic improvement techniques [13]. Currently, there is a gap in exploring the effect of rhizosphere bacteria on drought tolerance in *M. ruthenica* (L.) based on 16S RNA amplicon sequencing technology.

Rhizosphere microbiome maintains a symbiotic relationship with plant growth and development. Soil microbiomes that inhabit and symbiose in the rhizosphere of many plants exert a variety of beneficial effects on host plants through different mechanisms, such as nitrogen fixation and nodulation [14]. At the same time, they can also improve plant growth through siderophore production, the dissolution of insoluble phosphates, and the release of hormones [15]. *Aspergillus flavus* CHS1 could significantly increase the chlorophyll content, root–shoot length, and biomass yield of soybean plants under NaCl stress by regulating endogenous plant hormone levels and antioxidant enzyme activities up to 400 mM [16]. They have also shown that the soil microbiome can effectively increase the nutrient content of crops and improve the quality and quantity of crops by improving soil properties and secreting enzymes [17]. Yu [18] et al. significantly increased plant biomass, stem diameter, and plant height by inoculating plants with plant growth-promoting rhizobacteria (PGPR), while the presence of AMF promoted root length and the number of root shoots and reduced root volume, mean diameter, and the root–crown ratio. Singh [19] et al. studied the structural and functional changes in the root microbiome of 10 plant species of indica rice varieties and found that the α and β diversity indices of the rhizosphere microbiome with host genotypes revealed the changes in the structure of the root microbiome as well as the strong correlation with the host genotypes.

In this study, we analyzed the differences in the rhizosphere soil microbial diversity of different *M. ruthenica* (L.) varieties and their changing rules by examining the growth, physiology, and rhizosphere microbial changes of three varieties of *M. ruthenica* (L.) under drought stress, revealing the relationship between plant response to drought and the role of key bacteria and providing a theoretical basis for screening and constructing drought-resistant flora of *M. ruthenica* (L.) and its dry cultivation.

## 2. Materials and Methods

### 2.1. Plant Materials and Soil Properties

The three varieties of *M. ruthenica* (L.) selected for this experiment were all independently researched and bred by Inner Mongolia Agricultural University. The three varieties were *Medicago ruthenicus* ‘Zhilixing’ (ZL), *Medicago ruthenicus* ‘Mengnong No. 2’ (M2), and *Medicago ruthenicus* ‘Mengnong No. 1’ (M1). The soil used in this experiment was collected from the experimental base of Inner Mongolia Agricultural University (Hohhot, China) (111°22′30″ E, 40°41′30″ N), and the soil, which had not yet been planted or fertilized, was transported to the laboratory to be naturally air-dried, and then large debris and weed seeds were removed by passing it through a 2 mm mesh sieve [20]. It was stored at 4 °C. The soil had a pH value of 8.27, an organic carbon content of 17.25 g/kg, a fast-acting phosphorus content of 276.98 mg/kg, a fast-acting potassium content of 11.46 mg/kg, an ammonium nitrogen content of 0.91 mg/kg and a nitrate nitrogen content of 1.17 mg/kg.

### 2.2. Plant Germination, Transplanting, Cultivation, and Drought Stress Treatment

Seeds of uniform size and fullness were polished with sandpaper to break the hard solid surface, and then surface was sterilized in an ultra-clean table with 4% NaClO solution for 5 min, followed by rinsing with sterile water. The sterilized seeds were placed in sterile petri dishes, inverted in 4 °C dark environment culture for two days, then in 25 °C dark environment culture for two days, and finally in 25 °C light environment culture for one day. Soil substrate was prepared using sterilized vermiculite, soil, and perlite; mixed at a volume ratio of 6:3:1; and placed in 10 × 10 cm pots, and then 1/2 Hoagland nutrient solution [21] to saturation water content was added. The uniform seedlings were transplanted into pots: 5 plants were planted in each pot, 12 pots were cultivated for each material, and then 6 pots of blank soil (BS) were set up as the control. The seedlings were cultured in an artificial climate chamber at light/dark (16/8 h) and day/night temperatures (25 °C/22 °C) with a relative humidity of about 60% and were rehydrated to 70% water content every 3 d with sterile water. After two weeks, the seedlings were subjected to drought treatment. We controlled the soil moisture content by weighing and supplementing it with sterile water every day. The control group (N) and drought stress group (D) maintained soil moisture content at 70% and 30%, respectively, resulting in a total of 8 treatments, including M2N, M1N, ZLN, M2D, M1D, ZLD, BSN, and BSD. The drought stress lasted for 21 d.

### 2.3. Sample Collection

During the experiment, we measured plant height every 4 d. At d 21, single plants were sampled for aboveground and belowground biomass, and inter-root soil and blank soil were collected. We would cut the plants from the base and immediately weigh the aboveground portion of the plants for fresh weight and then dry them in an oven at 65 °C for 48 h, followed by determination of dry weight. Three replicates of each sample were taken, and their non-rhizosphere soil blank controls were collected. The method of soil and root separation was based on [22] and is briefly summarized as follows: the soil in the pots was snapped onto clean sterilized filter paper, the roots were pinched and gently shaken to remove the excess soil, placed into 50 mL centrifuge tubes, and phosphate buffer was added to the centrifuge tubes (6.33 g NaH_2_PO_4_-H_2_O, 16.5g Na_2_HPO_4_-7H_2_O, and 200 μL Silwet L-77 in 1l ddH_2_O, pH = 7.0). This was followed by short vortexing on an oscillating vortex mixer, removal of the roots and placing them in 15 m tubes to await subsequent use, and then filtration of the buffer solution in the tubes through a 100 μm filtration suspension cell filter. Finally, centrifugation on a centrifuge at 4000× *g* rpm for 15 min occurred, the supernatant was poured off, and the resulting precipitate was defined as rhizosphere soil, frozen in liquid nitrogen, and stored at −80 °C.

### 2.4. Measurement Indicators and Methods

For plant height, we measure from the base of the bud to the highest leaf at the tip of the plant. For growth rate, we used plant height/day. Fresh weight and dry weight were weighed using a ten-thousandths balance [23]. The root–crown ratio was calculated using root dry weight/aboveground biomass dry weight.

Proline (Pro) content was determined using the ninhydrin solution color development method [24]. Firstly, 0.5 g of each of the fresh leaves of the plants to be tested in different treatments were accurately weighed and placed in large tubes, and then 5 mL of 3% sulphosalicylic acid solution was added to each tube, extracted in a boiling water bath for 10 min, then filtered in clean test tubes after cooling, and the filtrate was the extract of proline. Subsequently, 2 mL of the extract was sucked into another clean test tube with a glass stopper, 2 mL of glacial acetic acid and 2 mL of acidic ninhydrin reagent were added, and the solution was heated in a boiling water bath for 30 min; the solution was red. After cooling, add 4 mL of toluene, shake for 30 s, let stand for a moment, take the upper layer of liquid to 10 mL centrifugal tube, centrifuge at 3000 r/min for 5 min, and then gently suck up the upper layer of proline red toluene solution in the colorimetric cup, with toluene as the blank control. With the spectrophotometer at 520 nm wavelength of the color, absorbance values can be obtained.

The high content of malondialdehyde (MDA) indicates a high degree of peroxidation of plant cytoplasm and serious damage to the cell membrane. We used the thiobarbituric acid method [25], weighing 0.2 g of each of the fresh leaves of the plants to be tested from different treatments and grinding them with 0.05 mol/L phosphate buffer at pH 7.8 in an ice bath to obtain 10 mL of homogenate. Transfer to a centrifuge tube and centrifuge at 13,000 r/min for 10 min to obtain malondialdehyde extract. A total of 2 mL of malondialdehyde extract was aspirated, and 2 mL of 0.5% thiobarbituric acid (TBA, prepared with 10% trichloroacetic acid) solution was added. The homogenate reacted in a boiling water bath for 15 min and then quickly cooled down and was centrifuged at 4000 rpm for 5 min. The supernatant was taken to determine the optical density values at 532 nm and 600 nm, and 0.5% thiobarbituric acid solution was used as the reference solution.

About the determination of catalase (CAT) [26], we took 2.5 g of the fresh leaves of the plants to be tested from different treatments and added a small amount of phosphate buffer solution of pH 7.8, ground it into a homogenate, transferred it to a 25 mL volumetric flask, rinsed the mortar with this buffer and transferred the rinsing solution to a volumetric flask, and fixed it with the same buffer and centrifuged it for 15 min at 4000 r/min. The supernatant was the crude extract of catalase.

### 2.5. DNA Extraction, PCR Amplification, Library Construction and Sequencing

The genomic DNA of the samples was extracted by the CTAB method, and the purity and concentration of the extracted DNA were detected by agarose gel electrophoresis; then, an appropriate amount of DNA was taken in a centrifuge tube and diluted to 1 ng/μL in a sterile water. PCR amplification of the 16S V3-4 region using barcoded-specific primers (515F and 806R primers: 5′-CCTAYGGGRBGCASCAG-3′, 5′-GGACTACNNGGGTATCTAAT-3′) using the diluted genomic DNA as a template.

PCR products were detected by agarose gel electrophoresis at a concentration of 2%, and aliquots were mixed according to the concentration of PCR products. After sufficient mixing, they were detected again using agarose gel electrophoresis at a concentration of 2%, and the destination bands were recovered using a gel recovery kit provided by Qiagen.

The library was constructed using NEBNext^®^ Ultra™ IIDNA Library Prep Kit, and the constructed library was subjected to Qubit and Q-PCR quantification; after the library was qualified, NovaSeq6000 was used for on-line sequencing. Each sample data were separated from the off-line data according to the barcode sequence and PCR amplification primer sequences, and the reads of the samples were spliced using FLASH (V1.2.11, http://ccb.jhu.edu/software/FLASH/, 25 November 2022) software to obtain the Raw Tags after truncation of the barcode and the primer sequences. The Raw Tags were then quality controlled using fastp software to obtain high-quality Clean Tags, and finally, the Clean Tags were compared with the database using Vsearch software to detect chimeras and remove them to obtain the final effective data, i.e., the Effective Tags. For the Effective Tags obtained above, the DADA2 module or deblur in QIIME2 software was used for noise reduction (DADA2 was used by default), and sequences with an abundance of less than 5 were filtered out to obtain the final ASVs (Amplicon Sequence Variants) and feature list. Subsequently, the ASVs were compared with the database using the classify-sklearn module of the QIIME2 software to obtain species information for each ASV.

### 2.6. Data Analysis and Bioinformatics Analysis

Statistical and bioinformatics analyses were performed using SPSS software (v.26.0.0.2). Analysis of variance (ANOVA) (Duncan’s multiple range test) was used for significance analysis between different groups (* *p* < 0.05, ** *p* < 0.01, *** *p* < 0.001, and **** *p* < 0.0001 were considered significant). Bar graphs were plotted using Prism 9.5.1 (GraphPad Prism). The alpha diversity of each sample was analyzed based on three alpha diversity indices: Chao1, Shannon, and Simpson. NMDS analyses based on weighted_unifrac distance were performed using R. Linear discriminant analysis of effect size (Lefse) was statistically significant (*p* < 0.05) with a log LDA score set at 2.0, and the analyses of all the above metrics were performed on the NovoMagic cloud platform.

## 3. Results

### 3.1. Phenotyping

Throughout the experimental period, we recorded the growth of three varieties of *M. ruthenica* (L.) under normal watering and drought stress conditions (Figure 1). In the plant height increment under the two treatments for each variety of *M. ruthenica* (L.), we can see (Figure 2a) that the increment under normal watering was M1 > M2 > ZL, while under drought stress on *M. ruthenica* (L.) the increment was M2 > M1 > ZL. The plant height increment of all three varieties of *M. ruthenica* (L.) shrank to varying degrees after experiencing drought stress, with the post-stress M2, M1, and ZL reduced by 24.86%, 34.37%, and 31.97%, respectively, which showed that M1 had a greater change in plant height increment after experiencing drought stress, while M2 was less affected by drought stress.

In the aboveground biomass fresh weight and dry weight plots of plants (Figure 2b), we can see that the aboveground biomass size under normal watering and drought stress were M2 > M1 > ZL; at the same time, the fresh weight and dry weight of the three varieties of *M. ruthenica* (L.) also showed different degrees of reduction due to drought stress, which was shown that the fresh weight of M2, M1, and ZL affected by drought decreased by 39.19%, 50.22%, 41.12%, while dry weight decreased by 23.26%, 26.10%, 24.49%. This indicated that it was M1 that was more affected by drought stress, while it was M2 that was less affected by drought, which suggests that M2 has a better ability to accumulate aboveground biomass in the face of drought stress.

Looking at the plant root–crown ratio (Figure 2c), we can see that there was no significant difference in the root–crown ratio among all varieties of *M. ruthenica* (L.) under both normal watering and drought stress. Under normal watering, the root–crown ratios of the three varieties of *M. ruthenica* (L.) did not differ significantly, with the smallest root–crown ratio being that of M1, which may be due to its low accumulation of aboveground biomass material. Under drought stress, M1 had the largest root–crown ratio, which may be due to its aboveground part development being better than the other two types of *M. ruthenica* (L.) under drought conditions.

### 3.2. Physiological and Biochemical Analyses

There were changes in the Pro of three varieties of *M. ruthenica* (L.) at the late stage of drought stress treatment (Figure 3a). Under normal watering, there was no significant difference in the Pro content of each variety of *M. ruthenica* (L.). But, under drought stress, the proline content of all three varieties of *M. ruthenica* (L.) showed a significant upward trend, with M2 having the highest content, followed by M1, and ZL had the lowest content. And we found that the Pro content of both M2 and M1 differed from that of ZL in a highly significant way. This shows that ZL has the worst drought resistance among the three varieties of *M. ruthenica* (L.), and we suspect that M2 is more drought resistant.

We can see the changes in the MDA content of each variety of *M. ruthenica* (L.) after experiencing drought stress (Figure 3b). Between the different varieties of *M. ruthenica* (L.), the MDA content was always the least in M2. After undergoing drought stress, the MDA content of all varieties of *M. ruthenica* (L.) increased substantially, as shown by ZL > M1 > M2, and the MDA content of M2 was highly significantly different from that of M1 and ZL. This represents that after experiencing drought stress, M2 plant cells were the least traumatized, while it was ZL that suffered the most damage from drought-affected cells.

The changes in the CAT content of each variety of *M. ruthenica* (L.) after being subjected to drought stress can also be visualized (Figure 3c). When not subjected to drought stress, the CAT content of the three varieties of *M. ruthenica* (L.) varieties showed a flat state. When subjected to drought stress, the CAT content of all varieties of *M. ruthenica* (L.) decreased to different degrees, with ZL showing the greatest change and the lowest CAT content among the three varieties, and the CAT content of ZL was highly significantly different from that of M2 and M1 under drought stress. The most minor change after experiencing drought stress was in M2, which had the highest CAT content among the three varieties of *M. ruthenica* (L.). This indicates that M2 is more antioxidant than M1 and ZL and has a higher ability to survive under drought conditions.

### 3.3. 16S RNA Bioinformatics Analysis

#### 3.3.1. Abundance Analysis of Soil Bacterial Communities

Through Illumina Hiseq PE250 high-throughput sequencing technology, a total of 1,943,222 sequences were obtained, of which 1,884,039 were valid sequences. Through the analysis of the sample dilution curve (Figure 4a), we can find that the coverage of each sample library was above 99.9% and that the sample dilution curves all tended to be flat, indicating that the sequencing data in this study are reasonable and can accurately reflect the real information of soil bacterial community. Chao1 (Figure 4b), Shannon (Figure 4c), and Simpson (Figure 4d) indices were used to analyze the diversity of the rhizosphere bacterial community in *M. ruthenica* (L.) under different treatments. The results showed that BS showed significant differences with M2, M1, and ZL in the Chao1, Shannon, and Simpson index under normal watering and drought stress. At the same time, we can also intuitively see that the diversity of the rhizosphere soil microbiome is also different among different *M. ruthenica* (L.) varieties. In addition, the bacterial diversity of M2 changed little after drought stress, while M1 and ZL changed greatly after drought.

#### 3.3.2. Analysis of Differences in the Composition of Rhizosphere Bacterial Communities

We used different points to represent each sample separately, with different colors representing different subgroups, and the distance between the points indicated the degree of difference. When the stress was less than 0.2, it indicated that the NMDS analysis had a certain degree of reliability and that the closer the samples were to each other on the coordinate graph, the higher the similarity was. We chose the non-metric multidimensional calibration NMDS analysis to measure the coefficient of dissimilarity between the three samples under normal watering and under drought stress. Therefore, we can find that the point distance between M2 and M1 is closer between the three samples of the rhizosphere soil of the three varieties of *M. ruthenica* (L.) under normal watering (Figure 5a), indicating that the bacterial community composition of the rhizosphere soil of M2 and M1 *M. ruthenica* (L.) is similar, while the point distance between the sample ZL and the points of M2 and M1N is further away from the samples, indicating that the bacterial community structure of the rhizosphere soil of ZL and the rhizosphere soil of the *M. ruthenica* (L.) of M2 and M1 is more different. While under drought stress conditions (Figure 5b), the point distances between the three varieties of *M. ruthenica* (L.) were similar, indicating that the bacterial community structure composition of the rhizosphere soil of the three varieties of *M. ruthenica* (L.) was similar.

We can see from the Veen diagram (Figure 5c,d) that there were a total of 6450 bacterial species in the samples under normal watering conditions, of which 1511, 661, 729, and 1169 were unique to BS, M1, M2, and ZL. A total of 6146 bacterial species were found in the samples under drought stress, including 1514, 727, 783, and 892 bacterial species unique to BS, M1, M2, and ZL.

#### 3.3.3. Analysis of the Composition of the Rhizosphere Bacterial Community

Through sequencing, we can find the characteristics of soil bacterial community distribution at the phylum level. The community structure of rhizosphere soil bacteria of three varieties of *M. ruthenica* (L.) under normal watering and drought stress conditions had high diversity at the phylum level (Figure 6a); the top ten bacterial phyla with relative abundance of bacterial communities in each treatment were Proteobacteria, Actinobacteriota, Acidobacteriota, Thermoplasmatota, Bacteroidota, Gemmatimonadota, Chloroflexi, Firmicutes, Crenarchaeota, and Myxococcota, respectively. Analyzing the proportion of each phylum of soil bacteria in the rhizosphere of three varieties of *M. ruthenica* (L.), it was found that there were some differences in the abundance of bacterial communities at the phylum taxonomic level among the samples. The Proteobacteria phylum was the most abundant, accounting for 72.83%, 64.41%, 67.64%, 62.88%, 58.30%, 59.68%, 34.18%, and 32.21% of the M2N, M2D, M1N, M1D, ZLN, ZLD, BSN, and BSD samples, respectively. The second most abundant phylum was Actinobacteriota, which accounted for 8.10%, 14.26%, 11.04%, 17.15%, 11.70%, 19.11%, 24.43%, and 29.67% of the samples, respectively.

At the genus taxonomic level (Figure 6b), the top ten bacterial taxa in terms of relative abundance of bacterial communities in the rhizosphere soil bacteria of the three varieties of *M. ruthenica* (L.) under normal watering and under drought stress conditions were *Novosphingobium*, *Rhizobium*, *Pseudomonas*, and *Pseudarthrobacter*, respectively, *Vicinamibacteraceae*, *Sphingomonas*, *Ensifer*, *Lysobacter*, *MND1*, and *Ralstonia.* Among them, *Novosphingobium* were the dominant taxa in plant rhizosphere soils, with the following proportions in each plant rhizosphere soil sample: 9.74%, 9.44%, 12.61%, 11.60%, 7.79%, 10.53%. *Rhizobium* was the second most dominant group with the following values: 5.28%, 6.97%, 3.65%, 6.00%, 2.47%, 4.6%.

We selected the top 30 genera from all the samples to further investigate the compositional heatmap of the rhizosphere bacteria of *M. ruthenica* (L.) (Figure 6c), and it can be seen that there is almost no difference in the non-rhizosphere soils, but interestingly there is an opposite trend in the abundance of the non-rhizosphere soil bacteria and the rhizosphere soil bacteria of the plant under normal watering. Under normal watering conditions, *Sphingobium*, *Limnobacter*, *Ideonella*, *Sphingomonas*, *Noviherbaspirillum*, and *Devosia* bacteria were abundantly present in the plant rhizosphere soil. Conversely, under drought stress conditions, these bacteria were fewer in number; *Streptomyces*, *Pseudarthrobacter*, *Glycomyces*, *Ensifer*, *Rhizobium*, *Pseudomonas*, *Piscinibacter*, *Novosphingobium*, and *Variovorax* were relatively abundant. Under drought conditions, *Glycomyces*, *Pseudomonas*, and *Rhizobium* were more prominent in the rhizosphere soil bacteria of M2, while *Streptomyces*, *Pseudarthrobacter*, and *Variovorax* were more prominent in M1. In the ZL, the expression of *Pseudarthrobacter* and *Glycomyces* were more prominent in M1.

#### 3.3.4. Identification of Potential Biomarkers in Rhizosphere Soils

Although we observed the effects of drought stress and varietal differences on changes in microbiome abundance in the rhizosphere soil of *M. ruthenica* (L.) and found some abundant bacterial flora in the rhizosphere soil, biomarkers in the rhizosphere soil remain unidentified. Therefore, we attempted to identify bacteria as biomarkers by performing a linear discriminant analysis of effect sizes (LEfSe) to most plausibly explain the observed differences between drought treatments or varieties. In this study, bacteria with LDA scores >2 were identified as biomarkers. The results showed that at the genus level, *Steroidobacter*, *Piscinibacter*, and *Novosphingobium* were considered as potential biomarkers in the rhizosphere soils of ZL, M2, and M1, respectively, under normal watering conditions (Figure 7a). Conversely, after experiencing drought stress, *Caulobacter*, *Xanthomonas*, and Chloroplast had high LDA scores in ZL, M2, and M1 rhizosphere soils, respectively, suggesting their potential value as novel biomarkers for screening rhizosphere soils under drought conditions (Figure 7b).

## 4. Discussion

### 4.1. Differences in Growth Response of Materials of Different Genotypes of the Same Plant following Drought Stress

In this study, growth indices, physiological and biochemical indices, and the composition of rhizosphere soil bacteria were determined in three varieties of *M. ruthenica* (L.) under normal watering and drought stress (Figure 2, Figure 3, Figure 4, Figure 5, Figure 6 and Figure 7). Numerous studies have shown that changes in habitat will inevitably lead to changes in plant phenotypic values, and the characteristics of the changes can well reflect the effects of environmental factors or the adaptability of this plant to habitat changes [27,28,29]. In this experiment, three varieties of *M. ruthenica* (L.) showed a decrease in plant height and a decrease in fresh and dry weight after drought stress (Figure 1 and Figure 2a,b), which was similar to the results of a large number of previous studies. In other words, a certain degree of drought can have a negative impact on the phenotypic values of plants, and the impact of water scarcity is very large, which can reduce the ability of plants to accumulate organic matter and reduce production [30,31]. At the same time, we found that M2 had the least amount of plant height increment and fresh weight and dry weight reduction after experiencing drought stress, indicating that it has the strongest ability to adapt to drought stress aggression. The root–crown ratio in this experiment also showed an increasing trend with drought stress (Figure 2d), which may be because drought stress caused the aboveground part of the plant to lose weight due to lack of water and, at the same time, promoted the growth of the belowground part of the plant to better search for water sources, increasing the root–crown ratio. These results indicated that drought stress would cause plants to preferentially allocate photosynthetic products to roots, which had a great effect on promoting the growth of roots. In previous studies, such morphological characteristics have been suggested to improve the drought resistance of plants, which is conducive to their timely response to drought stress [32]. At the same time, the three varieties of the *M. ruthenica* (L.) species showed different magnitudes of changes in the physiological indicators (Figure 3) when they were simultaneously increased or decreased, which reflects that drought stress triggers different plant responses depending on the plant genotype [33].

### 4.2. Differences in the Physiological Response of Different Genotypic Materials of the Same Plant Species after Drought Stress

When experiencing drought stress, plants adapt to drought not only by altering their epimorphology but also their osmoregulation, as well as increasing their activity of antioxidant enzymes. These are important ways for plants to cope with drought [34]. The results of this study showed that both Pro and CAT, excepting MDA, showed an increasing trend under drought stress (Figure 3), which indicated that *M. ruthenica* (L.) can resist damage by increasing their osmoregulatory substance content and antioxidant enzyme activities under drought stress [35,36,37]. At the same time, we can find that before being subjected to drought stress, there was no large Pro and CAT content in the three varieties of *M. ruthenica* (L.) difference, but after drought stress, M2 had the highest Pro and CAT content. Pro has the ability to scavenge reactive oxygen species; CAT can effectively scavenge hydrogen peroxide in plants and catalyze the decomposition of H_2_O_2_, which is a key functioning enzyme for plants to protect themselves from the toxicity of H_2_O_2_, indicating that its ability to resist drought stress was stronger. The more MDA accumulates in the plant, the more it can reflect the degree of damage brought by environmental stress to the plant [38]; this experiment also showed that M2 had the least amount of MDA accumulation and had a highly significant difference with M1 and ZL. It can also be seen that M2 has an excellent antioxidant capacity, which may evince the better drought resistance ability of M2.

### 4.3. Differences in Rhizosphere Bacteria of Different Genotypic Materials of the Same Plant Species after Drought Stress

The diversity of the bacterial community in the non-rhizosphere soil was significantly different from the diversity of the bacteria in the inter-rhizosphere soil of each variety of *M. ruthenica* (L.) (Figure 4, Figure 5c,d and Figure 6a,b). At the same time, the experimental results showed that the inter-rhizosphere soil recruited bacteria from the non-rhizosphere soil and that this recruitment was affected by both drought and plant genotypes. Indeed, the non-rhizosphere soil determines the microbial reservoir in the rhizosphere soil. The rhizosphere selects its microbiome from the non-rhizosphere soil, after which the plant selectively recruits certain bacteria to colonize its rhizosphere [39]. Aira M. [40] et al. found significant differences in the abundance and diversity of bacteria in the rhizosphere of maize of type su1 and type sh2. In the present experiment, it can be seen that the rhizosphere soil bacterial community composition analysis of M2, M1, and ZL *M. ruthenica* (L.) had significant differences in their rhizosphere microbial abundance and diversity (Figure 4, Figure 5 and Figure 6), which suggests that even for the same plant species, the different genotypes create significant differences in the microbial community. This suggests that species-driven bacteria are selectively recruited to the rhizosphere surface to interact with plants [41].

At the same time, it can be seen that at the phylum level (Figure 6a), drought can negatively affect the abundance of Proteobacteria in plant rhizosphere soils, resulting in a downward trend in the abundance of Proteobacteria. However, it is worth noting that the first type of bacteria gathered in the rhizosphere soil of all varieties of *M. ruthenica* (L.) was Proteobacteria, regardless of normal watering or drought stress. These results indicated that Proteobacteria was the dominant flora in the rhizosphere bacteria of *M. ruthenica* (L.), which was contrary to Wang K. et al. [42]. It may be due to the different core bacteria recruited by different plants. In non-rhizosphere soils, it was found that this phylum was not affected by drought stress and did not change much. For the Actinobacteriota phylum, drought brings positive effects, which led to an increase in the abundance of Actinobacteriota phylum in the rhizosphere soil of plants after drought stress, which suggests that drought stress environment is more favorable for the survival of Actinobacteriota phylum, and Actinobacteria have better drought resistance. Also, they can reduce the harmful effects of drought on plants [43]. At the genus level (Figure 6b), we can see that the abundance of *Novosphingobium* decreases in M2 and M1 but increases in ZL, reflecting the different genotypes of the same plant to recruit the bacterium under drought stress. At the same time, the abundance of *Rhizobium* in the rhizosphere soil of each variety of *M. ruthenica* (L.) showed an increasing trend after drought stress, indicating that *Rhizobium* was more conducive to colonization in an arid environment and helped plants adapt to arid conditions. *Rhizobium* has been found to increase the growth rate of alfalfa [44], and it is worth noting that the nitrogen-fixing bacterium *Rhizobium* was present in large quantities in the M2 rhizosphere soil under normal watering, which may be one of the reasons why the M2 was able to better accumulate aboveground biomass under drought conditions. It has been shown that *Novosphingobium* is a genus that is enriched every time alfalfa is subjected to drought stress, and it has been speculated to be the core bacteria of alfalfa [45], whereas in this study, we found that *Rhizobium* was enriched in large quantities in all varieties of *M. ruthenica* (L.) after they were subjected to drought stress, and it might be considered as the core bacteria associated with *M. ruthenica* (L.). Combined with the results of this experiment, we can clearly understand (Figure 7) that the core bacteria recruited by different plants vary greatly, and even the rhizosphere-specific bacteria driven to be recruited by different varieties of the same plant are different.

### 4.4. Rhizosphere Bacteria’s Growth and Drought Tolerance

It is worth noting (Figure 4b–d and Figure 5c,d) that the bacterial diversity of Rhizosphere soil in M1 and ZL decreased after drought stress, and only the value of M2 remained stable. However, the diversity index of the soil bacterial community in the rhizosphere decreased to 92.92% and 82.27%, respectively, due to drought stress in M2 and ZL. We can guess that the rhizosphere soil of M2 has more bacteria suitable for drought conditions. Moreover, after drought stress, the differences between the soil bacterial communities of the three kinds of *M. ruthenica* (L.) narrowed (Figure 5a,b and Figure 6) and tended to be similar. This may be because the drought stress eliminated some soil bacteria that were not suitable for drought conditions and left those that were adapted to drought conditions, which resulted in the similarity of the soil bacterial structure among the three varieties of *M. ruthenica* (L.) rhizosphere. At present, it seems that M2 has a more stable ability to recruit specific bacteria, or it recruits mostly drought-resistant bacteria itself, which may be one of the reasons for the more outstanding growth and drought-resistant ability of the M2.

The rhizosphere is an important site where plants connect to the outside world in response to drought stress. Understanding the recruitment of the rhizosphere soil microbiome under drought conditions and identifying core taxa in rhizosphere soils provides an outstanding contribution to the exploration of potential mechanisms by which the rhizosphere enhances plant drought tolerance. Our study provides evidence that rhizosphere soils recruit bacteria from non-rhizosphere soils, that their microbial composition and abundance are regulated by drought stress and the plant host, and that the process of bacterial enrichment is indeed driven by the genotype of the plant.

## 5. Conclusions

We investigated the changes of rhizosphere bacteria in three varieties of *M. ruthenica* (L.) under drought stress and combined them with the phenotype and 16S high-throughput sequencing results. We screened a variety of *M. ruthenica* (L.) with outstanding drought resistance, Mengnong No. 2. The results showed that among the three varieties of *M. ruthenica* (L.), Mengnong No. 2, Mengnong No. 1, and Zhilixing, the increase in plant height decreased by 24.86%, 34.37%, and 31.97%, the fresh weight decreased by 39.19%, 50.22% and 41.12%, and the dry weight decreased by 23.26%, 26.10%, and 24.49%, respectively, after being affected by drought stress. The state of Mengnong No.2 was also optimal in physiological tests. We found that there were some differences in the population structure of soil bacteria in the rhizosphere of different varieties of *M. ruthenica* (L.). At the phylum level, Proteobacteria was the most abundant. At the genus level, Novosphingobium belonged to the dominant group. We also selected the biomarkers of Mengnong No.2, Mengnong No. 1, and Zhilixing under drought stress, Xanthomonas, Chloroplast, and Caulobacter, respectively. Finally, our experiments showed that drought stress had a greater impact on the composition of the rhizosphere microbial community of *M. ruthenica* (L.) and that its physiological and biochemical reactions and bacterial enrichment processes were clearly driven by plant genotype. The drought-tolerant cultivar Mengnong No. 2 already had a high abundance of drought-resistant bacteria in the rhizosphere under the condition of sufficient water, while the other two varieties with poor drought tolerance could accumulate a certain abundance of drought-resistant bacteria in the rhizosphere after drought stress. Therefore, drought-tolerant varieties can respond effectively and quickly to drought stress, while varieties with poor drought-stress tolerance need to spend more time adapting to the drought environment, which affects their growth and yield. In this way, beneficial strains can be isolated to improve the drought resistance of *M. ruthenica* (L.).

From the aspect of the interaction between soil microbiome and plants, we have laid an important foundation for exploring the mechanism of plant resistance to drought stress, the screening of drought-resistant bacteria, and the construction of microbiota.

## Figures and Tables

**Figure 1 microorganisms-11-02851-f001:**
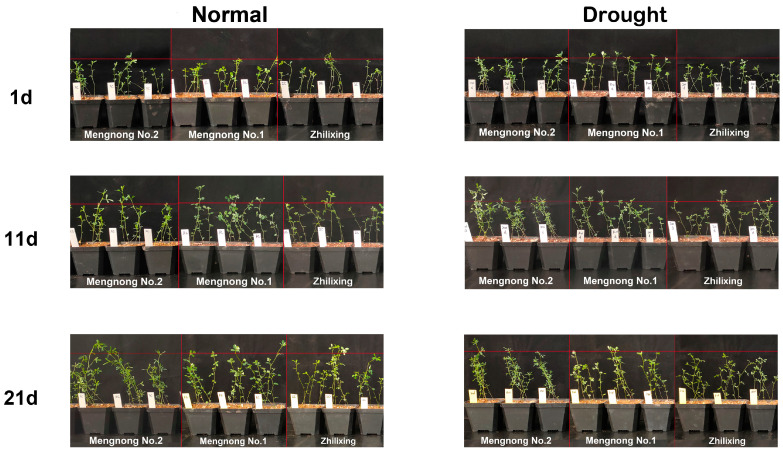
Photographs of growth phenotypes of three varieties of *M. ruthenica* (L.) (M2, M1, ZL) during drought stress.

**Figure 2 microorganisms-11-02851-f002:**
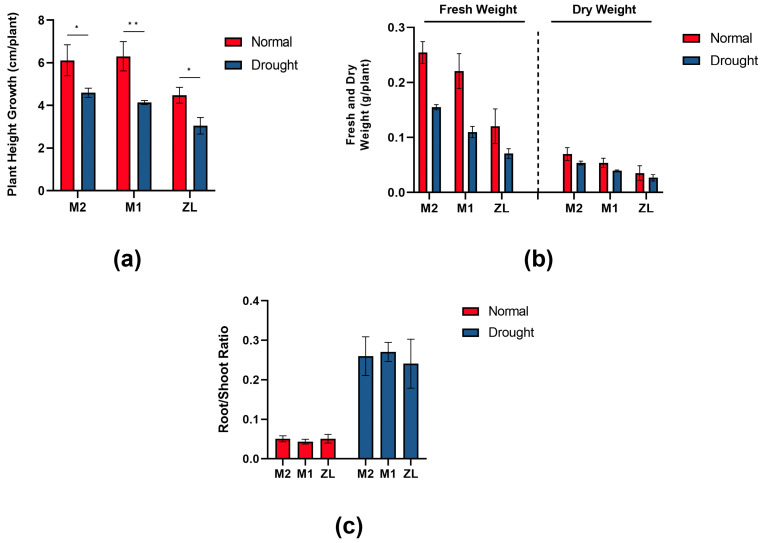
Changes in growth characteristics of three varieties of *M. ruthenica* (L.) during drought stress. (**a**) Plant height growth; (**b**) fresh and dry weight; (**c**) root–crown ratio. ANOVA analysis of variance was used to analyze the significance of the growth characteristics of the three varieties of *M. ruthenica* (L.) under normal watering and drought stress. Data are means of 15 replications, and error lines indicate standard deviation. Asterisks indicate statistical significance; * *p* < 0.05, ** *p* < 0.01 were considered significant.

**Figure 3 microorganisms-11-02851-f003:**
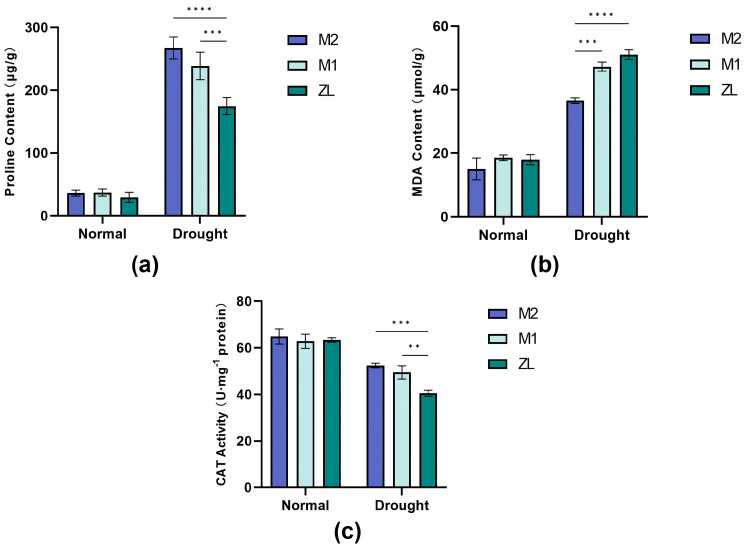
Changes in physiological characteristics of three varieties of *M. ruthenica* (L.) during drought stress. (**a**) Proline; (**b**) malondialdehyde; (**c**) catalase. ANOVA analysis of variance was used to analyze the significance of physiological characteristics of three varieties of *M. ruthenica* (L.) under normal watering and drought stress. Data are means of 15 replications, and error lines indicate standard deviation. Asterisks indicate statistical significance; ** *p* < 0.01, *** *p* < 0.001, and **** *p* < 0.0001 were considered significant.

**Figure 4 microorganisms-11-02851-f004:**
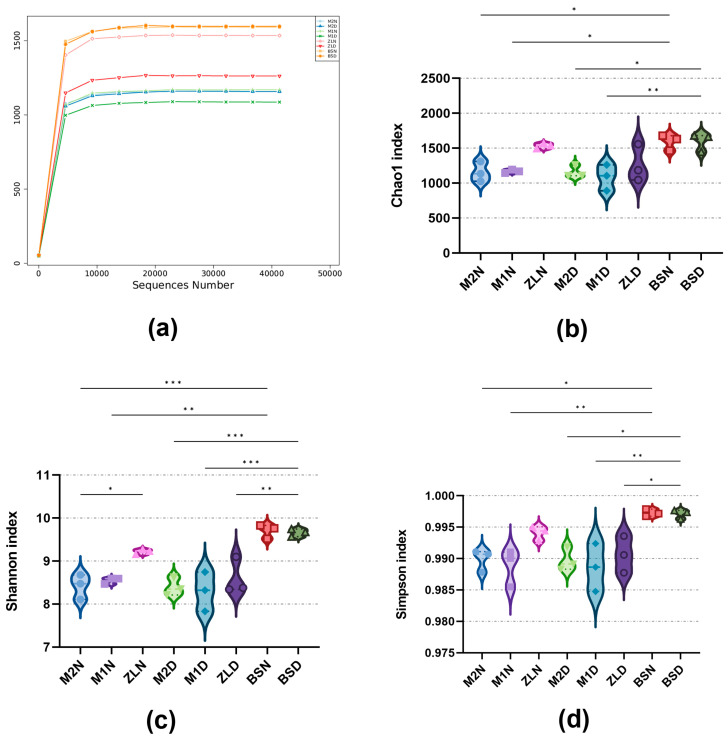
Changes in rhizosphere bacteria of three varieties of *M. ruthenica* (L.) under normal watering and drought stress. (**a**) Sample dilution curve; (**b**) Chao1 index; (**c**) Shannon index; (**d**) Simpson index. Asterisks indicate statistical significance; * *p* < 0.05, ** *p* < 0.01, *** *p* < 0.001, are considered significant.

**Figure 5 microorganisms-11-02851-f005:**
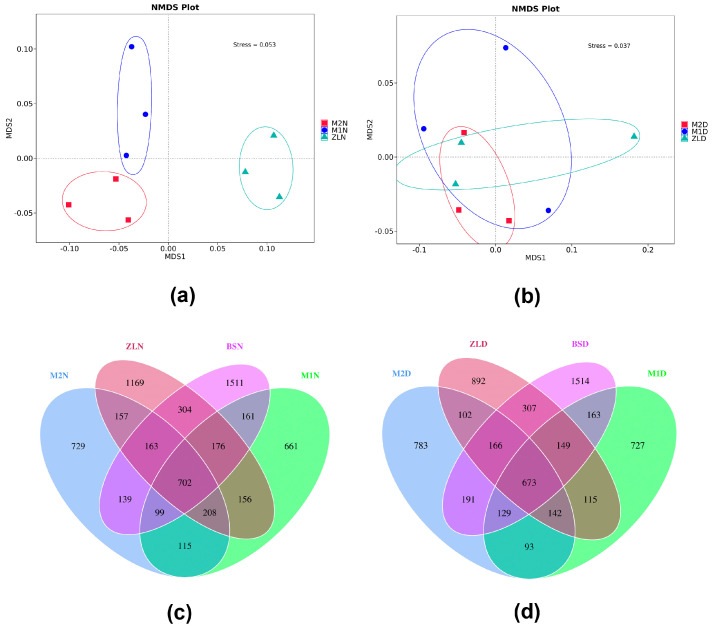
Differential analyses of microbial diversity based on different water treatments and varietal differences. (**a**) NMDS analysis based on weighted_unifrac distance for three varieties of *M. ruthenica* (L.) under normal watering conditions; (**b**) NMDS analysis based on weighted_unifrac distance for three varieties of *M. ruthenica* (L.) under drought stress conditions; (**c**) umber of overlapping and different OUTs in rhizosphere and non-rhizosphere soil (BS) samples of M2, M1, and ZL under normal watering conditions. (**d**) Overlap and number of different OUTs in the rhizosphere and non-rhizosphere soil (BS) samples of M2, M1, and ZL under drought stress.

**Figure 6 microorganisms-11-02851-f006:**
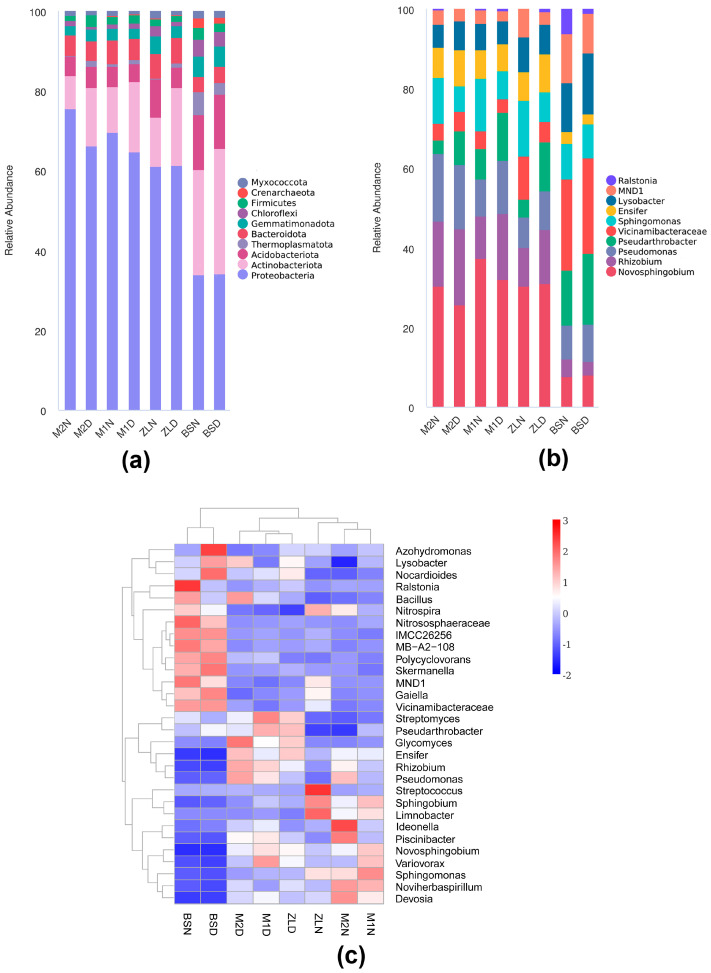
Differences in the composition of rhizosphere soil bacterial communities of three species under different water conditions treatments. (**a**) The percentage of the bacterial community (relative abundance top 10) on the phylum level in different water and varietal treatments; (**b**) the percentage of the bacterial community (relative abundance top 10) on the genus level in different water and varietal treatments; (**c**) a heat map clustering shows the average relative abundance of the top 30 genera of all samples.

**Figure 7 microorganisms-11-02851-f007:**
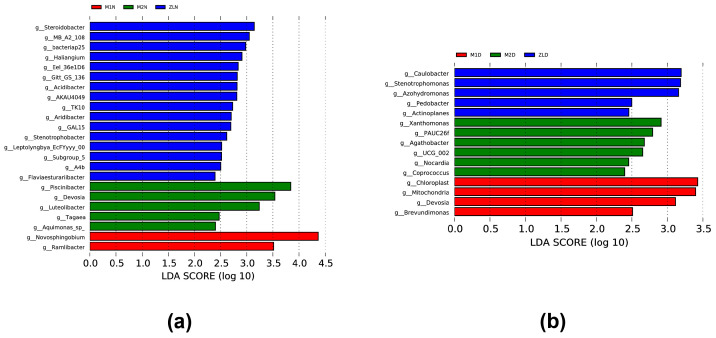
Potential biomarkers identified in rhizosphere soil. (**a**) Potential biomarkers at normal watering at the genus level; (**b**) potential biomarkers of drought stress at the genus level. LEfSe showing the differential abundance of genera in the rhizosphere soil of the three varieties of *M. ruthenica* (L.) species for normal watering and drought treatments based on a critical value of *p* < 0.05 and an LDA score of >2.0. The length of the bars indicates the effect of the corresponding genus in each group. LDA, linear discriminant analysis.

## Data Availability

The datasets presented in this study can be found in online repositories. The names of the repository/repositories and accession number(s) can be found at: https://www.ncbi.nlm.nih.gov/, 5 April 2024, PRJNA1023673 (SAMN37668277-SAMN37668300).

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
