# Peer review of "Variety-Driven Effect of Rhizosphere Microbial-Specific Recruitment on Drought Tolerance of Medicago ruthenica (L.)"

_microorganisms, 2023, doi:10.3390/microorganisms11122851_

Round 1
Reviewer 1 Report
Comments and Suggestions for Authors
This manuscript is titled “Variety-driven effect of rhizosphere microbial specific recruitment on drought tolerance of M. ruthenica (L.)”. However, for greater clarity it would be greatly appreciated if the full scientific name of Medicago ruthenica L. is displayed. The authors affiliations (line 5-6)should be numbered in accordance with their display in authors names.
Abstract
Line 12: This scientific name should be written in full for the first mention.
Line 12-13: …..but it is still unclear what the key role of microbes in resisting drought stress is. This requires revision.
Line 13-14: Therefore, we selected three varieties of M. ruthenica (L.) for drought treatment. Are these varieties selected for drought tolerance?
Introduction
Line 44: Medicago ruthenica should be in italics. Please also verify if authority name L should be in brackets.
Line 71: Delete "A" after Singh and " include year of publication.
Line 106: what is the full name of BS?
Materials and methods
Line 197: This should be written as : Analysis of variance (ANOVA).......
Results
Figure 1: The components/ blocks of this figure should be aligned and should be equal in size.
Figure 2: Figure 2 (d) is missing but the caption is included in this figure.
Please ensure that all the genera of bacterial species isolated in this study are italicized. For examples, Line 357-359; 369; 371-375; etc
Check consistency. In line 392 LDA is written in abbreviation but in line 396 it written in full with abbreviation in brackets.
References
References 35 and 36 should be in line with the rest of the cited authors. The authors names should not be written in capital letters.

Author Response
请参阅附件。

Reviewer 2 Report
Comments and Suggestions for Authors
The manuscript ID microorganisms-2687612 entitled " Variety-driven effect of rhizosphere microbial specific recruitment on drought tolerance of M. ruthenica (L.) is an interesting study. But my suggest it needs to a major revision to be published in this journal. I do have some comments about the manuscript and data interpretation/discussion that could improve the overall quality of the manuscript:
1- The authors repeated the term microorganisms in title, abstract, key words introduction and conclusion . I would only like to ask the authors why this term was referred to in this manuscript? I suggest to change this word by bacteria ( manuscript focus only on bacteria).
2- Abstract :
Line 12: M. ruthenica must be written as complete for first time.
3- Introduction : With the length of the introduction and the large number of references (19), the authors neglected many points, including the importance of plant growth prompting bacteria and its mechanisms to induce the plant growth and tolerance to biotic and a biotic stresses . ( I suggest these related and racent references to help the authors and improve the introduction… https://doi.org/10.3390/metabo11070428, https://doi.org/10.3390/jof8080775, https://doi.org/10.1007/s12010-022-03975-9, 10.21608/EJCHEM.2022.124008.5532, https://doi.org/10.1007/s40858-022-00544-7, https://doi.org/10.1007/s13399-023-03949-9, https://doi.org/10.1007/s13399-023-04510-4, https://doi.org/10.15835/nbha51313302).
4- Materials and methods: Wonderfully written. But there is some minor suggestions:
(lines 92:95) I suggest to transferee to the results section.
Line 100: "for 2d" change to " two days ).
Line 137 : leaves: what the types of leaf (dry or fresh).
Line 151: leaves: what the types of leaf (dry or fresh).
Lines 160:162: " Catalase (CAT) can effectively scavenge hydrogen peroxide in plants and catalyze the decomposition of H202, which is a key functioning enzyme for plants to protect themselves from the toxicity of H202 [26]." I suggest transferee to introduction section or to discussion.
Line 162 : leaves: what the types of leaf (dry or fresh).
5- Results :
Figure 2, 3 and 6 : needs more clarity. I suggest that separate the figures below each other.
Figures 4, 5, and 6 are complex and difficult to understand and require an explanatory explanation. However, in this way, they need a statistician to understand them.
6- Discussion : Wonderfully written.
7- Conclusion section: Comprehensive and explains what has been achieved in the study.
8- References: The reference list contains seven self-citation references, which is a high percentage compared to the total number of references 45. ( see 4,6,9,10,35,42,45).
Round 2
Reviewer 2 Report
Comments and Suggestions for Authors
The manuscript ID microorganisms-2687612 entitled " Variety-driven effect of rhizosphere microbial specific recruitment on drought tolerance of M. ruthenica (L.) The authors have made the required modifications and the manuscript has become excellent and suitable for publication in this journal, so I recommend publishing it. I thank you very much for your patience and effort in improving this manuscript.